# Exploring Frailty in the Intersection of Cardiovascular Disease and Cancer in Older People

**DOI:** 10.3390/ijerph20237105

**Published:** 2023-11-23

**Authors:** Erkihun Amsalu, Ying Zhang, Christopher Harrison, Tan Van Nguyen, Tu Ngoc Nguyen

**Affiliations:** 1Westmead Applied Research Centre, Sydney Medical School, Faculty of Medicine and Health, University of Sydney, Sydney, NSW 2145, Australia; erkihun.amsalu@sydney.edu.au; 2School of Public Health, Faculty of Medicine and Health, University of Sydney, Sydney, NSW 2050, Australia; ying.zhang@sydney.edu.au (Y.Z.); christopher.harrison@sydney.edu.au (C.H.); 3Department of Interventional Cardiology, Thong Nhat Hospital, Ho Chi Minh City 70000, Vietnam; nguyenvtan10@ump.edu.vn; 4Department of Geriatrics & Gerontology, University of Medicine and Pharmacy, Ho Chi Minh City 70000, Vietnam

**Keywords:** frailty, cardiovascular disease, cancer, cardio-oncology, multimorbidity

## Abstract

Advances in cardiovascular therapies and cancer treatments have resulted in longer patient survival. The coexistence of cancer and cardiovascular disease has been recognized as a complex clinical scenario. In addition to cardiovascular disease, older people with cancer are at greater risk of experiencing multimorbidity and geriatric syndromes, such as frailty. In older people, the concurrent presence of cancer and cardiovascular disease increases the risk of mortality, and the presence of frailty can exacerbate their conditions and hinder treatment effectiveness. Given the significant intersection among frailty, cardiovascular disease, and cancer in older people, this paper aims to provide an overview of the current research in this field and identifies gaps in the research to understand the burden and impact of frailty in these populations. While many studies have examined the prevalence and impact of frailty on adverse outcomes in patients with cancer or cardiovascular disease, evidence of frailty in individuals with both conditions is lacking. There is no universally accepted definition of frailty, which leads to inconsistencies in identifying and measuring frailty in older adults with cardiovascular disease and cancer. The frailty index seems to be a preferred frailty definition in studies of patients with cancer, while the frailty phenotype seems to be more commonly used in cardiovascular research. However, differences in how the frailty index was categorized and in how patients were classified as ‘frail’ depending on the cut points may have a negative effect on understanding the impact of frailty in the studied populations. This makes it challenging to compare findings across different studies and limits our understanding of the prevalence and impact of frailty in these populations. Addressing these research gaps will contribute to our understanding of the burden of frailty in older people with cardiovascular disease and cancer, and improve clinical care protocols in this vulnerable population.

## 1. Introduction

Cardiovascular disease and cancer are the two leading causes of mortality worldwide. According to the World Health Organization’s recent report, cardiovascular disease accounted for the most deaths related to non-communicable diseases (approximately 17.9 million people annually), followed by cancers (9.3 million), chronic respiratory diseases (4.1 million), and diabetes (2.0 million) [1]. In 2020, approximately 19.3 million people were newly diagnosed with cancer globally, with breast cancer being the most commonly diagnosed cancer, followed by lung, colorectal, prostate, and stomach cancers [2]. Advances in cardiovascular therapies and cancer treatments result in longer patient survival, and the number of cancer survivors aged 65 years or older has significantly increased [3]. Due to increased life expectancy, the chronic disease burden in older people with cancer is predicted to substantially grow over the next generation [4]. The coexistence of cancer and cardiovascular disease has been recognized as a complex clinical scenario [5]. In addition to cardiovascular disease, older people with cancer are at greater risk of experiencing multimorbidity and geriatric syndromes such as malnutrition, sarcopenia, and frailty, and these conditions can increase adverse outcomes, including all-cause mortality and cardiovascular mortality [6].

In recent years, there has been an increasing number of studies examining frailty in older patients with cardiovascular disease, as well as in patients with cancer. In older people, the concurrent presence of cancer and cardiovascular disease increases the risk of mortality, and the presence of frailty can exacerbate their conditions and hinder treatment effectiveness. Given the significant intersection among frailty, cardiovascular disease, and cancer in older people, this paper aims to provide an overview of the current research in this field to understand the burden and impact of frailty in these populations and to identify any gaps in this research.

## 2. Advancement in Frailty Research

Frailty is a state of increased vulnerability to stressors, resulting from decreased reserve and resilience across multiple physiological systems [7,8]. The concept of frailty was first described in the literature in the 1950s [9]. Since then, more than 40 frailty definitions have been proposed [9,10,11]. However, there is limited consensus on the “gold standard” to define frailty and measuring this complex syndrome presents challenges both for research and clinical practice. The two most widely accepted approaches to measuring frailty are the frailty phenotype and the cumulative deficit models [8,12]. The frailty phenotype (also called Fried’s frailty criteria) defines frailty with five criteria (unintentional weight loss, weak grip strength, self-reported exhaustion, slow gait speed, and low physical activity), and the presence of three or more of these five components is defined as being frail [13]. The cumulative deficit model forms the basis of Rockwood and Mitnitski’s Frailty Index, which is based on different age-related deficits in which frailty is a continuous score, considering signs, symptoms, functional disabilities, and chronic health conditions [14]. In addition to these two approaches, the Clinical Frailty Scale (CFS), the Edmonton Frail Scale, the Reported Edmonton Frail Scale, and the FRAIL scale are also commonly used to assess frailty in older people [12]. In addition, there are many rapid screening tools for frailty, such as PRISMA-7, the Tilburg Frailty Index, the Vulnerable Elders Survey, the Self-Rated Health Deficits Index, the Sherbrooke Postal Questionnaire, the G8 Questionnaire for oncology patients, the Easy Care short version, the Study of Osteoporotic Fractures (SOF) Index, and the Identification of Seniors at Risk (ISAR) [12]. The Asia-Pacific Clinical Practice Guidelines for the Management of Frailty recommends that frailty should be routinely screened for in adults aged 70 years and older, or those who had an unintentional loss of more than 5% of their body weight in the previous year [12]. This guideline also suggests that, when making the decision regarding a suitable frailty measurement, it is crucial to opt for one that not only effectively identifies frailty and predicts patient outcomes, but is also user-friendly, extensively validated, and aligned with the priorities, resources, and objectives of the particular clinical setting [12].

The prevalence and incidence of frailty vary depending on the measurement tools used, study populations, and study settings. In older community-dwelling people, the prevalence of frailty is approximately 10% [15]. In a recent meta-analysis of 240 population-level studies reporting 265 prevalence estimates from 62 countries and territories (with 1,755,497 participants), the pooled prevalence in studies using the physical frailty phenotype definition was 12% (95% confidence interval 11–13%), compared with 24% (95% confidence interval 22–26%) in studies using the deficit accumulation model [16]. In another systematic review and meta-analysis involving data from 120,805 community-dwelling older adults from 28 countries, the incidence of frailty was estimated at 43.4 new cases per 1000 person-years [17]. In clinical settings and long-term care settings such as hospitals and nursing homes, the prevalence was reported to be higher [18]. A recent meta-analysis conducted by Boucher and colleagues on frailty prevalence and outcomes in unplanned hospital admissions showed that among 45 cohorts (with a total of 39,041,266 admissions, a median age of 80 years, and 22 frailty measurement tools) the overall prevalence of moderate and severe frailty ranged from 14.3% to 79.6% [19].

In older people, frailty is associated with an increased risk of adverse outcomes resulting from the interactions between existing frailty features and new stressors [8]. These events can lead to a decline in overall health status, a limiting of daily activities, a reduction of mobility, an increase in the risk of falls and disability, and a negative impact on social relationships [12,20]. Frailty also has adverse effects on quality of life and can contribute to depression, dementia, cognitive impairment, hospitalization, and mortality rates in older individuals [20]. The World Health Organization has identified frailty as a target for preventive interventions due to its association with adverse outcomes [7,21]. When effectively managed with appropriate interventions to address identified deficits, frailty can be reversible, potentially reversing negative consequences [22].

## 3. The Association of Frailty and Cardiovascular Diseases

Many studies have focused on frailty in patients with cardiovascular disease. In older people, cardiovascular disease is a known risk factor for frailty, and frailty is associated with worse health outcomes in people with cardiovascular disease [23]. Systemic inflammation and oxidative stress may contribute to the development of both cardiovascular disease and frailty [23]. Both cardiovascular disease and frailty are influenced by many shared risk factors, including low physical activity, smoking, dietary patterns, obesity, and diabetes [24].

In a recent review, Wleklik and colleagues summarized the evidence of frailty across a spectrum of cardiovascular disease, including in patients with coronary artery disease, heart failure, hypertension, atrial fibrillation, implantable devices, cardiac surgery, and transcatheter aortic valve replacement [24]. Frailty was present in 45% to 80% of patients with heart failure, 30% in patients with chronic coronary heart disease, 50% in patients with acute myocardial infarction, 14% in patients with hypertension, 6% to 75% in patients with atrial fibrillation, 13% in patients with an implantable device, 75% in patients qualified for implantable cardioverter-defibrillator (ICD) implantation, and approximately 50% in older patients undergoing cardiac surgeries [24].

In another review conducted by Marinus and colleagues in 30 studies comprising 96,841 participants, the authors examined the prevalence of frailty in patients aged 60 years or older with cardiovascular disease and assessed mortality rates in patients with frailty and cardiovascular disease [25]. The review revealed that frailty was present in up to 75% of patients with heart failure, 12% to 74% of patients with aortic stenosis undergoing transcatheter/surgical aortic valve replacement or transcatheter aortic valve replacement (TAVI), 10% to 63% in patients with stable coronary heart disease, 27% to 48% in patients with acute coronary syndrome, 19% in patients undergoing percutaneous coronary intervention (PCI), 20% to 50% in patients undergoing coronary artery bypass surgery (CABG), 25% in patients suffering from cardiac arrhythmias, and 52% in patients undergoing aortic/lower limb arterial intervention. The review also found that frailty was associated with a 2.5- to 3.5-fold increase in all-cause mortality risk. Among the 30 studies in this review, up to 20 different frailty assessment tools were used, with Fried’s frailty phenotype being the most commonly use (in 15 out of 30 studies) [25].

Many other studies have found that frailty and pre-frailty could be associated with an increased risk of developing cardiovascular disease and an increased risk of cardiovascular mortality. In 2017, Veronese and colleagues conducted a meta-analysis and exploratory meta-regression analysis to examine the risk of cardiovascular disease morbidity and mortality in frail and pre-frail older adults [26]. They identified 18 cohort studies with a total of 31,343 participants. After a median follow-up of 4.4 years, compared to the robust groups, there was an increased risk for a faster onset of cardiovascular disease in the frail (hazard ratio 1.70, 95% confidence interval 1.18–2.45) and pre-frail (hazard ratio 1.23, 95% confidence interval 1.07–1.36). Compared to robust people, frail people had a 4-fold increased risk of cardiovascular mortality (hazard ratio 3.89, 95% confidence interval 2.39–6.34), and prefrail people had a 3-fold increased risk of cardiovascular mortality (hazard ratio 2.80, 95% confidence interval 1.83–4.28).

The connection between frailty and cardiovascular diseases could also be related to the under-prescription or over-prescription of cardiovascular therapies. Several studies have shown that inadequate blood pressure control or under-treatment of cardiovascular disease can worsen cardiovascular conditions in older people and hence increase the risk of frailty in this population [27,28,29]. On the other hand, there are potential risks related to the over-treatment of cardiovascular disease in older people. For example, excessive diuresis can lead to electrolyte imbalances, kidney injury, orthostatic hypotension, and falls in frail people [28,30].

These findings highlight that the prevalence of frailty is high among people with cardiovascular disease, and underscore the importance of identifying and managing frailty in this population to improve their health outcomes and overall well-being. 

## 4. The Association of Frailty and Cancer

Globally, nearly 20 million new cases of cancer are diagnosed each year, with approximately 60% of new cancer cases occurring in people aged 65 or above [2]. Frailty, comorbidity, and disability are common in older patients with cancer [31]. Studies in older adults with cancer have mentioned chemotherapy intolerance, postoperative complications, and death associated with frailty [31,32]. These highlight the crucial role of frailty assessment in making decisions regarding the management of older patients with cancer.

Frailty is common in patients with cancer. In a systematic review conducted by Handforth and colleagues in 2015 in 20 observational studies (with 2916 participants) with data on the prevalence and/or outcomes of frailty in older cancer patients with any stage of solid or hematological malignancy, the authors found that more than half of older patients with cancers had pre-frailty or frailty and those with frailty/prefrailty experienced a higher risk of chemotherapy intolerance, postoperative complications, and mortality [33]. The review showed that the reported median prevalence of frailty was 42% (range 6–86%) and of pre-frailty it was 43% (range 13–79%) [33]. In terms of risk prediction, frailty was independently associated with increased all-cause mortality (adjusted 5-year hazard ratio 1.87, 95% confidence interval 1.36–2.57), increased intolerance to cancer treatment (adjusted odds ratio 4.86, 95% confidence interval 2.19–10.78), and postoperative complications (adjusted 30-day hazard ratio 3.19, 95% confidence interval 1.68–6.04) [33]. Among the twenty studies in this review, six studies used the phenotype model to define frailty, while the rest used the cumulative deficit model [33]. In another review on the impact of frailty on health outcomes in older adults with lung cancer, the authors found that frailty had a strong and consistent association with mortality, with the adjusted hazard ratios for mortality in frail patients compared with fit patients ranged between 3.50 and 11.91 [34].

In contrast to research on frailty in people with cardiovascular disease where the frailty phenotype was commonly applied, the frailty index was more preferred in studies on frailty in patients with cancer. In a review conducted by Fletcher and colleagues, the authors aimed to explore how the frailty index (the cumulative deficit model) was applied to older adults with cancer. They found 41 studies, and there was significant variability in how these studies categorized the frailty index in older adults with cancer [35]. A cut-off point of 0.35 was most commonly used to categorize frailty. This cut-off value is higher than the cut-off point of 0.25 usually applied in studies in patients without cancer. The authors suggested that maintaining the frailty index as a continuous variable is likely to be beneficial until further validation studies determine optimum frailty index categories in this population.

There has been evidence that frailty can also predict the development of cancer. In a recently published paper, Mak and colleagues analyzed data from 453,144 UK Biobank participants and 36,888 Screening Across the Lifespan Twin study (SALT) participants, aged 38–73 years and with no cancer diagnosis at baseline [36]. They found a higher risk of any cancer in the frail compared to the non-frail UK Biobank participants (hazard ratio 1.22, 95% confidence interval 1.17–1.28 when defined by the frailty index, and 1.16, 95% confidence interval 1.11–1.21 when defined by the frailty phenotype), as well as in the SALT participants (hazard ratio 1.31, 95% confidence interval 1.15–1.49) [36].

In summary, studies have consistently shown that frailty is common in patients with cancer and is associated with a higher risk of treatment complications and mortality. Furthermore, frailty can also predict the development of cancer.

## 5. The Intricate Interplay between Cardiovascular Disease and Cancer

In recent years, significant attention has been directed toward understanding the interaction between cardiovascular disease and cancer, as their coexistence presents unique challenges in terms of diagnosis, treatment, and patient management. Cardiovascular disease and cancer share numerous modifiable and non-modifiable risk factors, including smoking, poor diet, excessive alcohol intake, obesity, physical inactivity, hypertension, dyslipidemia, aging, pollution exposure, and genetic predisposition [37,38] (Table 1). In patients with cancer, cardiovascular disease is a major cause of mortality and morbidity because of shared risk factors and the cardiotoxic effects of common anti-cancer treatments [39,40]. Several chemotherapy agents and radiation therapies used in cancer treatment can result in direct cardiovascular toxicity or exacerbate pre-existing cardiovascular conditions [41,42]. For example, anthracycline, a commonly used chemotherapeutic agent, can cause anthracycline-induced cardiotoxicity, which leads to changes in the heart’s structure and function, cardiomyopathy, heart failure, or arrhythmias [43]. Radiation therapy, particularly when delivered to the chest or mediastinum, can cause damage to the heart and blood vessels. This can result in conditions such as coronary artery disease, myocardial infarction, pericarditis, valvular disease, or cardiomyopathy [44]. Certain targeted therapy drugs, such as tyrosine kinase inhibitors (TKIs), may increase the risk of arterial thromboembolic events, venous thromboembolism, and pulmonary embolism [45,46]. It is important for healthcare professionals involved in cancer care to be aware of these potential complications and closely monitor patients for early detection and timely intervention to minimize their impact on patients’ cardiovascular health.

In addition, cancer can negatively impact cardiovascular treatment and outcomes. A recent systematic review of 21 studies reported that adherence to cardiovascular risk factor-related medications (such as glucose lowering therapies, statins, and antihypertensives) was low among cancer survivors and declined over time [47].

On the other hand, there has been evidence that the risk of cancer increases in adults with cardiovascular disease. In a study among 27,195,088 adults in the USA with a mean age of 43 and a median follow-up time of 33 months, those with cardiovascular disease were 13% more likely to develop cancer than those without cardiovascular disease (hazard ratio 1.13, 95% CI 1.12–1.13) [48]. Several medications used to treat cardiovascular disease may increase the risk of cancer over time (Table 1).
ijerph-20-07105-t001_Table 1Table 1The links between cancer and cardiovascular disease.Common risk factors [5,37,42]Aging, smoking, metabolic syndrome, low physical activity, poor diet, excessive alcohol intake, obesity, radiation, oxidative stress, air pollution, environmental toxins, genetic susceptibility.Effects of cancer treatment on the development of cardiovascular disease [5,42]Cardiac dysfunction and heart failure.Coronary artery disease, vascular toxicity and coronary spasms.Arrhythmias including atrioventricular block, prolonged QT interval, atrial fibrillation.Venous thromboembolism, peripheral vascular disease.Stroke.Hypertension.Reduced adherence to cardiovascular treatment in cancer survivors [47]Glucose-lowering therapies.Lipid-lowering therapies.Blood-pressure-lowering therapies.Common cardiovascular drugs that may increase cancer risk Diuretics: may increase risk of breast cancer, skin cancer, urinary cancer [49,50].Calcium channel blockers: may increase risk of skin cancer, urinary cancer, lung cancer, prostate cancer [51,52].Angiotensin-converting enzyme inhibitors (ACE inhibitors): may increase risk of lung cancer [53].

## 6. Limited Evidence of the Burden of Frailty in People with Concomitant Cardiovascular Disease and Cancer

Up to 20–30% of patients with cancer have pre-existing cardiovascular disease, while others may develop cardiovascular complications during or after cancer treatment [41]. While many studies have examined the prevalence and impact of frailty on adverse outcomes in patients with cancer or cardiovascular disease, evidence of frailty in individuals with both conditions is lacking.

In 2022, the European Society of Cardiology (ESC) Guideline on cardio-oncology, developed in collaboration with the European Hematology Association (EHA), the European Society for Therapeutic Radiology and Oncology (ESTRO), and the International Cardio-Oncology Society (IC-OS), was released. The role of frailty in the management of patients with cancer and cardiovascular disease was acknowledged several times in this guideline [41]. The guideline mentioned that the management of acute coronary syndrome in patients with cancer can be challenging because of frailty, increased bleeding risk, thrombocytopenia, and increased thrombotic risk [54]. Cardiac surgery is usually difficult in patients with cancer because of the presence of frailty and multimorbidity [41]. The guideline also identified several gaps in evidence in this field, including raising awareness of the benefits of minimizing cardiovascular risk in patients with cancer in order to reduce the risk of cancer therapy-related cardiovascular toxicity, and early diagnosis and treatment of cancer therapy-related cardiovascular toxicity to improve patient outcomes, quality of life, and frailty status, implying that frailty can be considered as an outcome indicator for research in cancer therapy-related cardiovascular toxicity [41]. However, there is very limited information on the epidemiology of frailty in the guideline.

Earlier this year, Cao and colleagues published a paper on the association of frailty with the incidence risk of cardiovascular disease in long-term cancer survivors, using data from the UK Biobank [55]. Their analysis of 13,388 long-term cancer survivors (median age 62 years and diagnosed with cancer over 5 years before enrolment; about 40% and 8% of participants were survivors of breast and colorectal cancer) without a history of cardiovascular disease showed that frailty was associated with an increased incidence risk of cardiovascular disease among long-term cancer survivors. Frailty was assessed using both the frailty phenotype and the frailty index at baseline. The incidences of cardiovascular disease and diabetes were obtained through linked hospital and primary care data. Over a median follow-up of 12 years, compared with non-frail participants, those with frailty had a significantly higher risk of cardiovascular disease: an adjusted hazard ratio of 2.12 (95% confidence interval 1.73–2.60) with frailty defined by the Fried’s frailty phenotype, and 2.19 (95% confidence interval 1.85–2.59) with frailty defined by the frailty index. A similar finding was obtained for pre-frailty: an adjusted hazard ratio of 1.18 (95% confidence interval 1.05–1.32) with frailty defined by the Fried’s frailty phenotype, and 1.51 (95% confidence interval 1.32–1.74) with frailty defined by the frailty index [55].

## 7. Discussion

Cancer and cardiovascular disease are prominent public health concerns that can create significant economic and social burdens. Patients with both cancer and cardiovascular disease face higher mortality rates, and the presence of frailty can worsen their conditions and impede their ability to undergo effective treatments. This area of research has received increased attention in recent years, and there are two major gaps in research regarding frailty in older people with cardiovascular disease and cancer that require further investigation.

Firstly, there is a lack of evidence on the prevalence and incidence of frailty in patients with both cancer and cardiovascular disease. Frailty in people with concomitant cardiovascular disease and cancer poses unique challenges and requires a comprehensive approach to care. Both cardiovascular disease and cancer are prevalent and interconnected health conditions, leading to an increased risk of frailty in affected individuals. Frailty is characterized by reduced physiological reserve and increased vulnerability to adverse health outcomes. The presence of both cardiovascular disease and cancer can exacerbate these vulnerabilities, making frailty management a crucial aspect of healthcare. Chemotherapy, radiation therapy, and surgical interventions, which are essential for cancer treatment, can impose significant physiological stress on the cardiovascular system, increase the risk of cardiac complications, including heart failure, arrhythmias, and myocardial infarction, and thereby accelerate frailty progression. Additionally, cancer-related treatments may cause damage to various organs, leading to functional decline and increased frailty.

Secondly, there is a lack of a standardized definition for frailty. There is no universally accepted definition of frailty, which leads to inconsistencies in identifying and measuring frailty in older adults with cardiovascular disease and cancer. The frailty index seems to be a preferred frailty definition in studies of patients with cancer, while the frailty phenotype seems to be more commonly used in cardiovascular research. However, differences in how the frailty index was categorized and in how patients were classified as ‘frail’ depending on the cut points may have a negative effect on understanding the impact of frailty in the studied populations. This makes it challenging to compare findings across different studies and limits our understanding of the prevalence and impact of frailty in these populations. As frailty exists on a spectrum, several researchers have preferred to consider it as a continuous variable, while many other researchers have categorized it into two groups (frail/non-frail) or three groups (fit or robust, prefrail, and frail) when reporting their study results.

Addressing these research gaps will contribute to our understanding of the burden of frailty in older people with cardiovascular disease and cancer, facilitating the development of prevention strategies, personalized interventions, and improved clinical care protocols in this vulnerable population. As the population ages and cancer treatments improve, an increasing number of older adults are surviving cancer and living with cardiovascular disease. However, this population is at a high risk of frailty, which can complicate cancer treatment and increase cardiovascular morbidity and mortality. Therefore, understanding frailty in older patients with cardiovascular disease and cancer is critical for providing the appropriate care and improving outcomes. In daily clinical practice, recognizing and assessing frailty in patients with CVD and cancer can guide treatment decisions and risk stratification, and aid in prognostic information and supportive care planning. It allows healthcare professionals to develop personalized care, optimize treatment strategies, and improve patient outcomes. Recognizing frailty early in the disease trajectory allows for appropriate interventions to mitigate its consequences. Identifying frailty can help clinicians determine appropriate cancer treatments and optimize cardiovascular management to improve the quality of life and survival rate. Interventions aimed at improving physical function, nutrition, and social support may help prevent or reverse frailty in these patients, reducing the burden of disease and improving their well-being [12,56]. Managing frailty in people with concomitant cardiovascular disease and cancer necessitates a multidisciplinary approach involving healthcare professionals from various specialties, including oncologists, cardiologists, geriatricians, rehabilitation specialists, general practitioners, and pharmacists. Comprehensive geriatric assessments should be conducted to evaluate patients’ functional status, nutritional status, cognitive function, and comorbidities. This evaluation can aid in identifying and addressing the physical, psychological, and social needs of patients, ensuring personalized and patient-centered care. The stages and types of cancer may be associated with frailty even before any antineoplastic therapies. Hence, frailty should be assessed at baseline with the cancer diagnosis, during treatment (e.g., chemotherapy or hormonal therapy), and after treatment. Assessing frailty at different stages of the cancer journey, including baseline, during treatment, and after treatment, is crucial for understanding its impact on patients’ health and well-being.

Although this review focuses on older people because of the age and co-existence of several other factors that put them at a high risk of developing frailty, it should be noted that younger patients with cancer are also at risk of developing frailty as a result of the disease itself. Thus, frailty assessment should also broaden to the wider population, including young patients with these conditions.

## 8. Conclusions and Future Directions

In conclusion, frailty in people with concomitant cardiovascular disease and cancer is a complex and multifactorial condition that requires further research. Early recognition of frailty through geriatric assessments, tailored interventions targeting physical and cognitive functioning, nutritional support, and psychosocial assistance can help improve the quality of life for these vulnerable individuals. A collaborative effort among healthcare professionals is essential to ensure optimal outcomes and minimize the impact of frailty on this unique patient population. It is worth noting that research in this area is still relatively new, and there is ongoing exploration to better understand the epidemiology, risk factors, and outcomes among individuals living with both cancer and cardiovascular disease.

Future research on frailty in older people with cancer and cardiovascular disease may focus on several areas to improve understanding, management, and outcomes. Other potential future directions include developing and validating standardized and practical tools to assess frailty, specifically in older individuals with concomitant cancer and cardiovascular disease. These tools should consider the unique characteristics and challenges of this population to accurately determine their frailty status. Secondly, further studies on biomarkers to identify reliable biomarkers or risk predictors associated with frailty in older patients with cancer and cardiovascular disease could aid in early identification, stratification, and targeted interventions to prevent or manage frailty in these populations. Thirdly, there should be randomized controlled trials to evaluate interventions aimed at preventing, reducing, or reversing frailty in this population. These interventions may include exercise and physical activity programs, nutritional support, medication management, and multidisciplinary care models. Assessing the impact of frailty on long-term outcomes, such as survival, quality of life, and functional independence, in older cancer patients with cardiovascular disease can help healthcare providers and patients make informed decisions about treatment options and supportive care. Finally, as mentioned above, there is a need to implement integrated care models that promote the collaboration of multidisciplinary teams to address the complex needs of older individuals with cancer and cardiovascular disease who are also frail. These models could help optimize treatment decisions, minimize treatment-related complications, and provide holistic care. Overall, future research on frailty in older people with cancer and cardiovascular disease will likely focus on personalized approaches, comprehensive assessments, and interdisciplinary collaborations to improve outcomes and enhance the overall care of these vulnerable populations. Future research is needed to explore the burden and impact of frailty in young patients with cancer or CVD.

## Data Availability

Data are contained within the article.

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
