# Peer review of "Exploring Frailty in the Intersection of Cardiovascular Disease and Cancer in Older People"

_ijerph, 2023, doi:10.3390/ijerph20237105_

Round 1

Reviewer 1 Report

Comments and Suggestions for Authors

I reviewed the manuscript about frailty in the intersection in CVD and canceer. It is an interesting topic with limited literature, that has been nicely analyzed.

some minor comments:

- frailty does not concern only older individuals, especially in cancer patients. So in discussion but also in conclusions and future directions, it would worth to broaden the population in which frailty should be assessed. In older indviduals because of the age and the co-existence of several other factors, frailty is expected to exist, but in cancer in younger patients frailty may be the result of the disease itself. This needs to be discussed further.

- other important issues that need to be highlighted are the fact that stage and type of the cancer may be associated with frailty even before any antineoplastic therapies. The type of therapies applied may also play a role, that is why frailty in cancer patients should be assessed at baseline with the cancer diagnosis, during treatment (eg chemotherapy or hormonal therapy etc) and after treatment.

- specific frailty assessment tools may need to be developed in patients with cancer and CVD and need to be validated in this population. 

Author Response

I reviewed the manuscript about frailty in the intersection in CVD and cancer. It is an interesting topic with limited literature, that has been nicely analysed.  

Response:  We would like to thank you for the time spent reviewing our manuscript and for the useful comments. We have specifically responded to the issues raised as detailed below.

1) Frailty does not concern only older individuals, especially in cancer patients. So, in discussion but also in conclusions and future directions, it would worth to broaden the population in which frailty should be assessed.  In older individuals because of the age and the co-existence of several other factors, frailty is expected to exist, but in cancer in younger patients frailty may be the result of the disease itself. This needs to be discussed further.    

Response:  Thank you for your constructive suggestions. We agree that frailty is not limited to older individuals and that younger cancer patients may also experience frailty as a result of the disease itself.  However, the focus of our paper is on older individuals, due to their high risk of frailty, multimorbidity and associated adverse outcomes. Taking into consideration your suggestion, we have added this paragraph to the Discussion part: 

“Although this review focuses on older people because of the age and co-existence of several other factors that put them at a high risk of developing frailty, it should be noticed that younger patients with cancer are also at risk of having frailty as a result of the disease itself. Thus, frailty assessment should also broaden to wider population including young patients with these conditions.” (lines 355-359)

 We have also added this sentence to the Conclusion and Future Directions part:

“Future research is needed to explore the burden and impact of frailty in young patients with cancer or CVD” (lines 394-395)

2) other important issues that need to be highlighted are the fact that stage and type of the cancer may be associated with frailty even before any antineoplastic therapies. The type of therapies applied may also play a role, that is why frailty in cancer patients should be assessed at baseline with the cancer diagnosis, during treatment (e.g. chemotherapy or hormonal therapy etc) and after treatment.   

Response: Thank you for your constructive suggestions and input. We agree that the stage and type of cancer may be associated with frailty even before any antineoplastic therapies.  We have added this paragraph to emphasize the importance of assessing frailty at baseline with the cancer diagnosis, during treatment, and after treatment in cancer patients:   

“The stages and types of cancer may be associated with frailty even before any antineoplastic therapies. Hence frailty should be assessed at baseline with the cancer diagnosis, during treatment (e.g. chemotherapy or hormonal therapy) and after treatment. Assessing frailty at different stages of the cancer journey, including baseline, during treatment, and after treatment, is crucial for understanding its impact on patients' health and well-being.” (lines 348-352)

3) specific frailty assessment tools may need to be developed in patients with cancer and CVD and need to be validated in this population.   

Response: We agree that specific frailty assessment tools may need to be developed and validated in patients with cancer and cardiovascular disease. We have added these sentences in the Conclusion and Future Directions part:  

“Other potential future directions include developing and validating standardized and practical tools to assess frailty specifically in older individuals with concomitant cancer and cardiovascular disease. These tools should consider the unique characteristics and challenges of this population to accurately determine their frailty status”. (lines 372-376)        

Reviewer 2 Report

Comments and Suggestions for Authors

The article describes important but vaguely defined clinical entity - frailty. The background is to cardiovascular disease and cancer. The review provides dataon the necessity on further research in the comorbid patients.  While the topic might be interesting, the authors have not provided ideas how to perform such studies, only general recomendations have been given.

Nevertheless the article is scientifically sound.

Comments on the Quality of English Language

Minor corrections only

Author Response

We would like to thank you for the time spent reviewing our manuscript and for the useful comments. We have specifically responded to the issues raised as detailed below:

1. The article describes important but vaguely defined clinical entity - frailty. The background is to cardiovascular disease and cancer. The review provides data on the necessity on further research in the comorbid patients.  While the topic might be interesting, the authors have not provided ideas how to perform such studies, only general recommendations have been given. Nevertheless, the article is scientifically sound. 

Response: Thank you for this suggestion and inputs. Regarding the vague definition of frailty, it was addressed in the introduction section of the manuscript. It acknowledged the lack of a universally accepted definition of frailty, which leads to inconsistencies in identifying and measuring frailty in older adults with cardiovascular disease and cancer. It mentions that the frailty index is commonly used in cancer studies, while the frailty phenotype is more commonly used in cardiovascular research. However, variations in categorization and classification of frailty based on cut-off points can affect the understanding of frailty's impact in different populations. This limitation is acknowledged in the manuscript.

We appreciate your comments and agree that frailty is a complex and multifactorial condition that requires further research.  While our review provides an overview of the current state of knowledge on frailty in older individuals with concomitant cancer and cardiovascular disease, we acknowledge that more specific recommendations on how to perform such studies would be beneficial. In this regard, the revised manuscript provides suggestions on how to perform future studies on frailty in this population (lines 371-395).

Reviewer 3 Report

Comments and Suggestions for Authors

The paper of Amsalu and coll. is interesting, however, some points should be clarified:

-       Definition of frailty. It could be useful for the readers a table summarizing the methods currently used to assess frailty. Reporting advantages and disadvantages of each method could be also useful.

-       The connection between frailty and cardiovascular diseases could be related also to the underuse of indicated class of drugs. The potential risk related to under prescription as well as those related to over prescription should be discussed more in detail

-       The relevance of the relationship described in the review for daily clinical practice should be better highlighted

Author Response

We would like to thank you for the time spent reviewing our manuscript and for the useful comments. We have specifically responded to the issues raised as detailed below.

1) Definition of frailty. It could be useful for the readers a table summarizing the methods currently used to assess frailty. Reporting advantages and disadvantages of each method could be also useful.

Response: Thank you for your suggestion. A summary of the methods currently used to assess frailty was present in the manuscript (lines 69-83). Advantages and disadvantages of each method have been discussed in many other review papers and is not a scope of this paper.

 2) The connection between frailty and cardiovascular diseases could be related also to the underuse of indicated class of drugs. The potential risk related to under-prescription as well as those related to over-prescription should be discussed more in detail.

 Response: We agree that the underuse or overuse of certain cardiovascular medications may also contribute to the connection between frailty and cardiovascular diseases.

We have added the following paragraph in the manuscript (lines 159-166):

“The connection between frailty and cardiovascular diseases could also be related to the under-prescription or over-prescription of cardiovascular therapies. Several studies showed that inadequate blood pressure control or under-treatment of cardiovascular disease can worsen cardiovascular conditions in older people and hence increase the risk of frailty in this population.[27-29] On the other hand, there are potential risks related to over-treatment of cardiovascular disease in older people. For example, excessive diuresis can lead to electrolyte imbalances, kidney injury, orthostatic hypotension, and falls in frail people.[28, 30]”

3) The relevance of the relationship described in the review for daily clinical practice should be better highlighted.   

Response: We have added this in the revised manuscript:  

“In daily clinical practice, recognizing and assessing frailty in patients with CVD and cancer can guide treatment decisions and risk stratification, and aid in prognostic information and supportive care planning. It allows healthcare professionals to develop personalized care, optimize treatment strategies, and improve patient outcomes.” (lines 331-335)

Round 2

Reviewer 3 Report

Comments and Suggestions for Authors

I've no further comment.